# Effects of Community-Based Interventions on Medication Adherence and Hospitalization for Elderly Patients with Type 2 Diabetes at Primary Care Clinics in South Korea

**DOI:** 10.3390/ijerph18073396

**Published:** 2021-03-25

**Authors:** Yoon-Joo Choi, Young-Taek Kim, Hyun-Suk Yi, Soon Young Lee, Weon-Young Lee

**Affiliations:** 1Department of Preventive Medicine, College of Medicine, Chung-Ang University, 84 Heukseok-ro, Dongjak-gu, Seoul 06974, Korea; luvya2054@naver.com; 2Public Health Medical Office, Chung-Nam National University Hospital, Daejeon 35015, Korea; ruyoung01@cnuh.co.kr; 3Gwangmyeong Public Health Center, Gwangmyeong-si, Gyeonggi-do 14303, Korea; yhs6294@korea.kr; 4Department of Preventive Medicine, College of Medicine, Ajou University, Suwon 16499, Korea

**Keywords:** type 2 diabetes mellitus, primary care, community

## Abstract

Korean Disease Control and Prevention Agency launched Control and Prevention Community-based Registration and Management for Hypertension and Diabetes mellitus Project (CRMHDP) in Gwangmyeong city, 2009. This project has provided incentives on both patient and physician and has made private clinics and Public Health Center (PHC) in a community collaborate for effective chronic disease management among elderly people. This study aimed to evaluate the effects of CRMHDP on medication compliance and hospitalization due to diabetes-specific complications. The retrospective cohort study design was based on data of Korean National Health Insurance (KNHI) with 2 control areas (A & B) with usual primary care service similar to Gwangmyeong city regarding community health resources. The data on the study subjects were examined for the following 5 years since the baseline point. Medication adherence rates of CRMHDP-enrollees after the project was significantly higher than two control groups. For the hospitalization due to any complications, adjusted hazard ratio in the intervention group, compared to the control group A and B, were 0.76 (95% Confidence Interval: 0.65–0.78) and 0.52 (95% Confidence Interval 0.41–0.78), respectively. CRMHDP could successful in improving the management of type 2 diabetes mellitus among elderly people in South Korean primary care settings.

## 1. Introduction

The global burden of diabetes had increased significantly since 1990. In 2017, global prevalence and death of diabetes mellitus 476.0 million and 1.37 million, with a projection 570.0 million and 1.59 million in 2025, respectively [1]. In South Korea, it was ranked as the sixth leading cause of deaths [2,3]. Early diagnosis and continuous treatment combined with changing health behaviors were emphasized to reduce the health and economic burden of diabetes. With continued, coordinated, and comprehensive primary care, health condition for the diabetes could be more easily managed [4]. Evidence demonstrated there were the needs to reorient primary care toward chronic condition [5,6].

The effective and efficient management for diabetes patient, however, has not yet been incorporated to primary care settings. For example, a primary care pay-for-performance scheme that rewards practices for delivering effective intervention in chronic conditions like the Quality and Outcome Framework of primary care in UK [7] and Comprehensive Primary Initiative in United States [8], have brought to only a limited quality improvement in chronic care management despite of a huge investment. One of reasons for the failure of chronic care model can be that primary care physicians are unable to put enough time in daily practice to treat patients with chronic disease due to administrative burdens [9].

According to Tricco et al. [10], patient-mediated quality improvement strategies like patient self-management education and reminder service were better than provider-centered interventions strategies emphasize audit and feedback, clinician education and financial incentives. In this regard, both physician-centered strategies without considerable work burden and patient mediated strategies would have been more effective in chronic care management in primary care settings. The Korean Centers for Disease Control and Prevention (KCDC) created a Community based Registration and Management for Hypertension and Diabetes mellitus Project (CRMHDP) in primary care settings for the elderly. The project began on 1 July 2009 in Gwangmyeong City. This project was designed by taking account of the socio-psychological and behavioral characteristics among the elderly and busy primary care environment in Korea. Since then, CRMHDP was rolled out to 19 areas, both urban and rural areas, and has been viewed as a role model to improve poor quality of Korea’s primary care system according to Organization for Economic Cooperation and Development in OECD report [11]. This study aimed to examine how CRMHDP in Gwangmyeong(GM) city, started in 2009, influenced the medication compliance as an evaluation indicator of intermediate outcome and hospitalization rates related to complications of type 2 diabetes mellitus (Type 2 DM) among elderly patients as an evaluation indicator of long-term outcome, comparing with control groups.

### 1.1. CRMHDP: Introduction Background of the Intervention

The Korean primary care system has unique characteristics. First, most of local clinics run by medical specialists including internal medicine, pediatrics, obstetrics & gynecology, otolaryngology, family medicine are solo practices with a private ownership and are reimbursed based on fee-for-service by National Health Insurance. The local clinics in the community are subject to Korean primary care system. It is more likely to encourage local physicians to lead more visits of patients to clinics. As a result, Korean local clinics have twice more visits from patients than the average visits in the OECD country [12]. Second, physicians in local clinics do not have enough time to provide patients with primary care services following the national clinical guidelines for hypertension and diabetes. Specialists were qualified to treating diabetes patients are internist and family doctor. The other specialists running local clinics, however, are allowed to treat patients with chronic disease including type 2 DM as well as diseases was dealt in their own field. Third, every city and counties in Korea have a public health center (PHC) operated by the local government whose chief role is to implement health promotion and education programs for the residents in the local government boundary. This project was developed for the private clinics to collaborate with the PHC in the community. A local PHC would educate patients on how to manage their conditions while private clinics registered diagnosed patients and provide needed treatment for the patients.

### 1.2. Underlying Theory and All Intervention Components

The chronic care model by Wagner highlighted the active participation of patient and provider to improve the health outcome during the chronic disease management [13]. This project included a plan for incentives to both elderly patients and physicians. Figure 1 showed the outlines of CRMHDP in Gwangmyeong comparing to the usual care service. Elderly patients with hypertension or type 2 DM aged 65 and over were registered to CRMHDP by local private clinics where they visited. They received three benefits. First, they were exempted from the co-payment (1.50 USD) for a clinic visit and from co-shared cost (2.00 USD) for the drugs insured by Korean National Health Insurance (KNHI) per a prescription. Second, appointment reminder services for scheduled clinic visits were provided to registered patients. If they did not visit the doctor for three months, the Registration and Education Center (REC) [14] at PHC with five staffs (a team leader, 2 nurses and 2 dieticians), made phone calls to check if there were problems. Third, participating physicians encourage the registered patients to attend two mass education sessions for self-management lasting one hour per a session at REC in the PHC. If a registered patient was required to receive intensive education sessions on diet, exercise, and diseases due to poor control of HbA1c, a personalized training and counseling session would be provided to the patient referred by physician at REC in the PHC. While this project did not offered a specified financial incentive to participating primary care physicians, doctors have participated and supported the project because it encouraged patients to keep their appointments with the registered clinics without shopping for the better clinic. Doctors were satisfied with more patients for their clinics as a result.

### 1.3. Illustration of Any Intended Interactions

Primary care-based registration supported by financial incentive made it possible to maintain a relationship between doctor and patient and to have longitudinal health records for quality of services and health outcomes. While the REC at a PHC provided education sessions and customized counseling sessions for elderly patients to be aware of problematic health behaviors and to be more confident how to manage their condition, primary care physicians could focused more on higher quality of health care without worrying about consultation with elderly patients.

## 2. Materials and Methods

We conducted a retrospective cohort study using controls from geographically matched areas with baselines 2009/2010 and followed up until 2014/2015; the follow-up period for participants was five years. Two cities of 26 cities in Gyeonggi provincial area to which GM city belongs were selected as the control groups areas (A & B) based on the similarity with GM city in area-level characteristic (SF1). The matched characteristics of control areas included population size, proportion of 65 aged and over in total population, financial autonomy of city governmental budget and the number of local clinics per 1000 people, portions of apartments in housing (SF1). In the control areas, primary care clinics provided usual care services including a short consultation and making a prescription without any incentive scheme like exemption of cost sharing on patient and mass education and personal counselling at PHC. The reason why this study used geographical matching areas to select control group was because the health of people with chronic disease could be influenced by neighborhood context. Among adults with chronic diseases such as diabetes, cardiovascular disease, and asthma, effective disease management often needs continuous clinical follow-up and lifestyle modification like diet and exercise which may be influenced by neighborhood environment such as available health care, accessible exercise facilities and available nutritious foods [15].

The study was based on the national health insurance data in agreement with the Korean National Health Insurance Service Act (KNHIS data base). Patient’s personal identification information was protected prior to analysis. KNHIS database, claimed by health care institute, contains all information regarding the health care use of all Korean people. In the KNHIS database, the identification of patients with type 2 diabetes mellitus is based on the presence of ICD-10 code E12 (type 2 diabetes mellitus) in diagnosis and the form of antidiabetic drugs (insulins, sulfonylureas, metformin, meglitinides, thiazolidinedione, dipeptidyl peptidase-4 inhibitors, and α-glucosidase).

Figure 2 showed the flowchart of type 2 DM patients considered for the study. The patients in the intervention group were patients with type 2 DM (*n* = 2416) aged 65 and over, who registered in the CRMHDP during baseline years at 72.1% (*n* = 75) of all local clinics (*n* = 104) except for three hospitals treated patients with type 2 DM in the GM city and did not have cancer or intractable disease as a comorbidity. Different type of medical specialists (Internist, family doctor, pediatrician, obstetrician & gynecologist etc.) ran the clinics participated in CRMHDP of which 38.3% were operated by internists or family doctors. The number of elderly patients participated in the intervention group accounted for 54.7% of all elderly patients with type 2 DM including undiagnosed patients in the city. Those who withdrew (*n* = 917) from the registration to the CRMHDP due to moving to clinics not registered in the same city during the follow-up period were excluded from the subject for the analysis, because they could not benefit from the registration to the CRMHDP. A total of 1499 patients, were included in the intervention group for the study, had regularly visited to a registered clinic until censored during the follow–up period. Control A group (*n* = 2279) at 86.7%(*n* = 77) of all local clinics (*n* = 89) except four hospitals treated type 2 DM patients in the area A and control B group (*n* = 1184) at 44 (88.3%) of all local clinics (*n* = 53) except for three hospitals treated the patients in the area B were included based on inclusion standard: patients aged 65 and over with type 2 DM. The portion of clinics run by internist or family doctor out of participating local clinics constituted 38.1% and 32.1% in control A and B area, respectively. The number of the elderly patients in the control group A and B constituted 63.7% and 71.4% of all elderly patients with type 2 DM including undiagnosed patients in each area, respectively. During the baseline years, researchers excluded the patients with cancer or intractable disease as a comorbidity. For the impact on the hospital admissions due to type 2 diabetes-specific complications, participants with complications such as myocardial infarction, stroke, renal failure, glomerular disorder in diabetes, arterial arteriolar and capillary disease, diabetic retinopathy, diabetic polyneuropathy and diabetes with ulcer during the baseline years were excluded from the statistical analysis in both intervention (*n* = 367) and control groups (*n* = 767 for group A, *n* = 396 for group B). The number of study participants for hospital admission was as follows: intervention group (1132), control A group (1512) and control B group (788).

### 2.1. End Points

Primary outcomes for the medical effectiveness of the CRMHDP were medication adherence to assess short-term effect and hospital admissions due to complications of type 2 diabetes for long-term assessment. The former was measured by Medication Possession Ratio (MPR) for antidiabetic drug or insulin injection, the number of days of prescription divided by days in a unit of time (e.g., 1 year). It should be noticed that MPR was assessed by prescribed usage instead of actual usage. For the long-term outcome, we analyzed the data on hospital admissions due to type 2 diabetes mellitus—specific complications including myocardial infarction, stroke, renal failure, glomerular disorder in diabetes, arterial arteriolar and capillary disease, diabetic retinopathy, diabetic polyneuropathy and diabetes with ulcer.

### 2.2. Statistical Analysis

The demographic and clinical characteristics including health care use prior to the initial participation of study were compared between the invention group and control groups. For each group, proportions of those who have more than 80% in MPR during the period of two years prior to and five years after initial registration were computed. A mixed analysis of variance was used to compare adherence rates between and within a paired two groups composed by the intervention group and a control group over the period of one year before the CRMHDP and two years after the project. This was adjusted for baselines covariates such as age, gender, household economic status, and comorbid complications of type 2 diabetes mellitus. Household economic status was measured by contributions of the insured made to the NHI.

Differences in hospital admissions due to type 2 diabetes-specific complications between the intervention group and control groups were examined using Kaplan Meier and a Cox proportional hazard regression model with adjustment for baseline covariates (age, gender, economic status, type of health care security, and medication compliance during two years prior to participation). An observation was treated as censored if the patient was dropped-out due to moving to another area or death during the study period. The observational periods for the CRMHDP participants and their matched controls were from the baseline period of 1 July 2009 to 31 December 2010 till 1 July 2014 to 21 December 2015. The results were presented as hazard ratios (HRs) with a 95% confidence interval (CI).

## 3. Results

Table 1 showed differences in sociodemographic and clinical characteristics in the intervention group, control group A and control group B. There were not significantly different in gender ratio but there were significantly different in age distribution. The intervention group (77.2%) and control group B (71.0%) had a higher proportion of the patients aged 65–74 than control group A (65.0%).

Income distribution quintiles were significantly disproportionate. While the proportion of individuals in the lowest quintile income group was higher in the intervention group than the two control groups, the proportion of the highest quintile income group was higher in the two control groups than the intervention group.

The proportion of patients with type 2 DM who had the average MPR of 80% during the period of two years before the project was significantly disproportionate among study groups. While the MPR in the intervention group was at 80.4%, the control group A and B were at 85.7% and 83.0%, respectively. The presence of type 2 diabetes-specific complications was significantly different among study groups. While the proportion of patients with any complication from type 2 DM was 24.5% in the intervention group, it was 33.4% and 33.6% in control A and B group, respectively.

Figure 3 and Figure 4 showed the comparison of trends of MPR among study groups before and after of participation in the CRMHDP. For the intervention group, their MPRs was increased by 12 percent one year after the participation and has sustained at such a level during the following five years after the participation in the CRMHDP. For both control A group and B group, MPRs of patients was increased by three or four percent and five or seven percent, respectively at one year after the study and continued at such a level during the following five years.

Table 2 showed the comparison of medication adherence rates between and within a paired group consisted of intervention group and control groups over the period of one year before the CRMHDP and two years after the project. For the intervention group and patients in the control A group the main effect of time was significant as adherence rate increased (F = 351.15, *p* < 0.001) and the interaction between time and group was significant (F = 13.31, *p* < 0.001). For the intervention group and control B group, the main effect of time was significantly positive (F = 298.73, *p* < 0.001) and the interaction between time and group had a significance at alpha error of one percent (F = 2.96, *p* = 0.081).

Figure 5 showed compared hospitalization within five years after CRMHDP enrollment due to diabetes-specific complications between intervention group and control groups in Kaplan-Meier curves. While the percentage of at least one of diabetic-specific complications in the past five years was 11.06% (125/1132) in the intervention group, the percentage in control group A and B was 15.86% (240/1512) and 21.08% (166/788), respectively.

Table 3 showed the adjusted hazard ratios of the intervention group comparing with the control groups for hospitalization due to type 2 diabetes-specific complications within five years after CRMHDP enrollment. For hospitalization due to any complications, adjusted HRs of the intervention group-comparing with control group A and B were 0.76 (95% CI: 0.65–0.91) and 0.52 (95% CI: 0.41–0.78), respectively.

## 4. Discussion

This study showed that medication adherence rates among those in CRMHDP more rapidly increased after the enrollment compared with the control groups and that the increased adherence rates among CRMHDP-enrollees were maintained during the follow-up period. The CRMHDP-enrollees were significantly less likely to be hospitalized by diabetes-specific complications than the control groups, after adjusting confounding variables like gender, age, economic status, health insurance type and medication adherence rates at baseline.

Some characteristic of CRMHDP were likely to produce these clinical effects. A small financial incentive for the patients registering to the project and visiting a clinic could result two benefits. The first benefit was stopping patients from shopping for physicians and clinics to have the same prescription for an identical health problem and then helped create trust between patient and physician over the long term [11]. The trust toward his/her physician could led to improving patient medication compliance [16]. According to Vlaev et al. [17], reducing cost sharing of health care use and medication could make patients with chronic diseases increase their medication compliance and improve their health outcome. Researchers found financial support were likely to be more effective in low-income groups [18]. Given that elderly Koreans still have the highest poverty rate among OECD members due to the lack of a basic pension fund [19], the exemption from out-of-pocket payments for health care and medication in primary care setting could help create desirable health behaviors from Korean elderly patients with chronic disease. Regular reminders of next appointment might help the elderly to come into a clinic continuously. Older people tend to rely less on deliberative capacity and more on intuition, rules of thumb and shortcuts [20]. According to a systematic review [21], recall and reminder messages could affect medication adherence for patients who needed a long-term medication. In the project, the enrolled patients missing the clinic visits for more than three months received calls and reminder messages from the REC at the PHC.

The second benefit was that most of local physicians in the city participated in the project and adequately played their roles, which were registration of patients and referring them to a REC at PHC if they needed education. In spite of primary care physicians’ heavy workload from numerous visits of patients to clinics, almost of them took active part in the project since it did not press an extra burden on them and could make them retain their patients like regular customers in Korean primary care setting where doctor shopping prevailed.

If the enrolled patients wanted to participate on a group education session for two hours concerning diet, exercise and general health awareness of diseases, they were referred to the REC at PHC by a corresponding physician. The education session took place at a café and community center near the clinics where participants consulted with their physicians. About 20% of total enrollees have received the education service during the five years after the registration project. If physicians encountered diabetes patients having hemoglobin A1c levels above 7%, they recommended personal training and education for self- management at the REC. This session ran for 40 to 50 min, with one session per week for six weeks. Approximately 50 patients participated on the session every year. These education service might make a limited contribution to producing the clinical effect of the registration project.

There were limitations to this study. First, the choice of control areas regardless of individual factors could have created biases. To modify the bias, we adjusted covariates (gender, age, income level, accompanying complications and medication adherence two years before starting research) in the statistical analysis. However, health behavior like diet control and regular exercise as potential confounders was not adjusted in the examination of differences in MPR and hospitalization by diabetic specific complication between intervention and control group.

Second, CRMHDP- enrollees would already be more competent than the control groups in self-management before the registration project. As shown Table 1, however, medication adherence rate in the intervention group was slightly lower than control groups before the project. Moreover, to prevent distortion of results from volunteer bias, medication compliance before starting research in both groups as a covariate were included.

Third, this study inevitably used prescribed day instead of actual usage as measuring MPR, because study data was extracted from National Health Insurance claimed data.

Volunteer bias and confounding in choosing the control areas, however, still could interfere with the true effect of the project on health outcome. A comparative study using controls selected by individual matching based on propensity score matching technique needs to be conducted in the future. Regardless of these limitations, this study showed that a chronic care for type 2 DM in primary care setting could be tailored to the primary care physicians and diabetes patients in the community.

Comparisons of geographical characteristics between Gwangmyeong city and two control group areas in 2009 see Appendix A.

## Figures and Tables

**Figure 1 ijerph-18-03396-f001:**
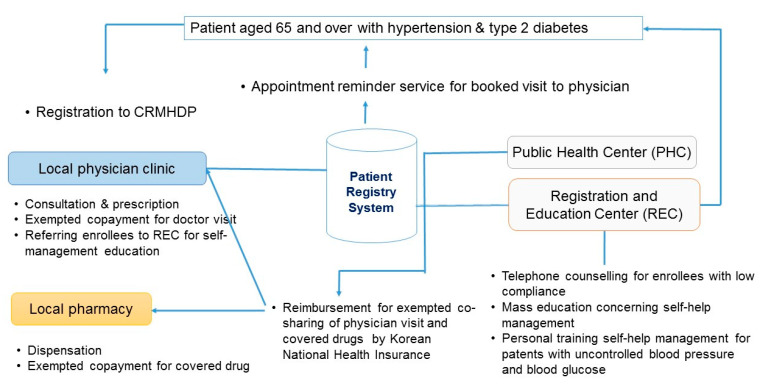
Outlines of CRMHDP in Gwangmyeong city.

**Figure 2 ijerph-18-03396-f002:**
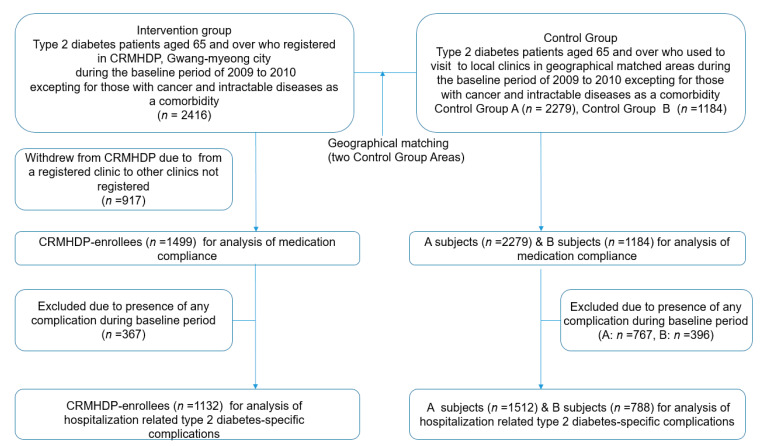
Flowchart of study subjects for analytic purposes in both Gwangmyeong (CRMHDP and Control group A and B.

**Figure 3 ijerph-18-03396-f003:**
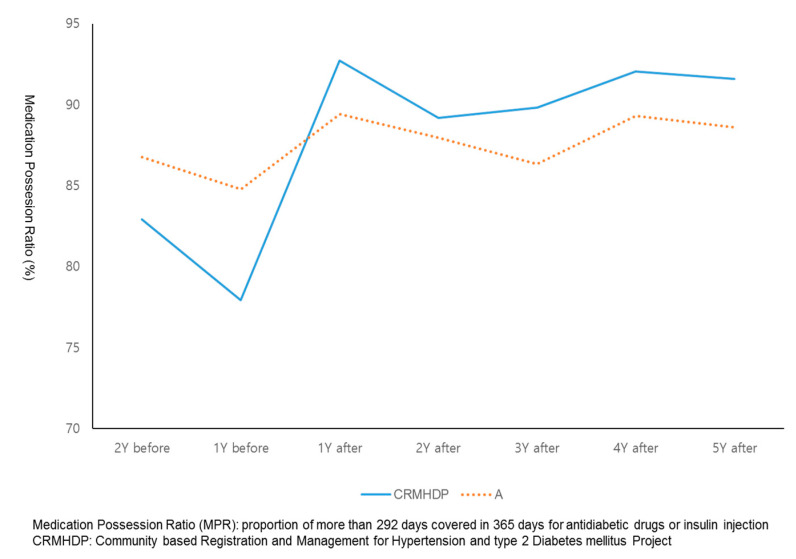
Comparison of trends of average MPRs between Gwangmyeong city (CRMHDP) and Control group A area before and after enrollment.

**Figure 4 ijerph-18-03396-f004:**
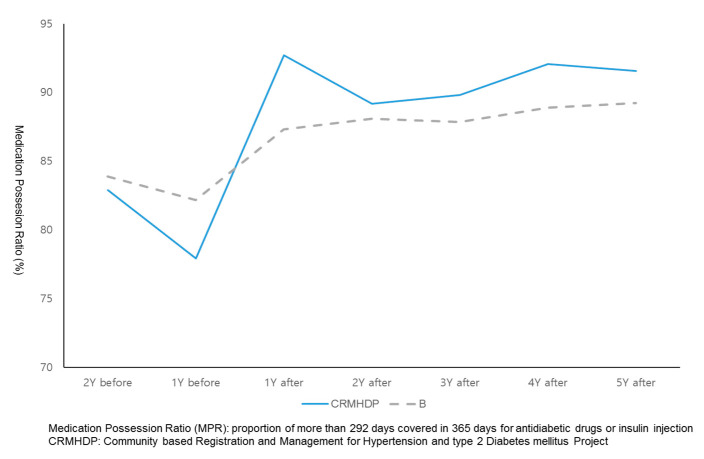
Comparison of trends of average MPRs between Gwangmyeong city (CRMHDP) and Control group B area before and after enrollment.

**Figure 5 ijerph-18-03396-f005:**
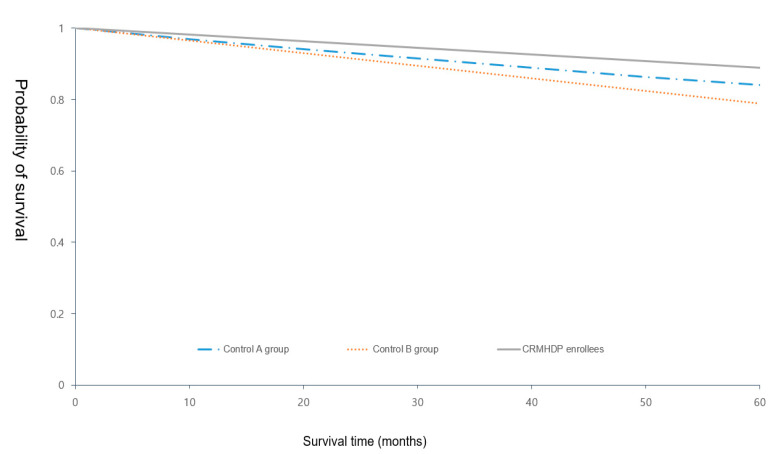
Kaplan-Meier curves for CRMHDP-enrollees and controls.

**Table 1 ijerph-18-03396-t001:** Comparisons of sociodemographic and clinical characteristics among study subjects by chi-square test.

Independent Variables		CRMHDP-Enrollees		Control A Group		Control B Group		χ^2^ *p*-Value
		n	(%)	n	(%)	n	(%)	
Age								72.66 (*p* < 0.001)
	65–74	1158	(77.2)	1481	(65.0)	841	(71.0)	
	75–84	323	(21.6)	714	(31.3)	312	(26.4)	
	85-	18	(1.2)	84	(3.7)	31	(2.6)	
Gender								2.7 (*p* = 0.258)
	Female	573	(38.2)	931	(40.9)	465	(39.3)	
	Male	926	(61.8)	1348	(59.1)	719	(60.7)	
Income Quintile								59.47 (*p* < 0.001)
	Q1 (Bottom 20%)	330	(22.0)	355	(15.6)	247	(20.9)	
	Q2	285	(19.0)	451	(19.8)	275	(23.2)	
	Q3	302	(20.1)	464	(20.3)	220	(18.5)	
	Q4	323	(21.5)	456	(20.2)	203	(17.1)	
	Q5 (Top 20%)	259	(17.2)	553	(24.3)	239	(20.2)	
Medical Possession Ratio ^1^ before starting research								18.7 (*p* < 0.001)
	80% or more	1205	(80.4)	1953	(85.7)	982	(83.0)	
	Less than 80%	294	(19.6)	326	(14.3)	202	(17.0)	
Presence of any complication ^2^ in baseline years								40.64 (*p* < 0.001)
	Yes	367	(24.5)	767	(33.6)	396	(33.4)	
	No	1132	(75.5)	1512	(66.4)	788	(66.6)	
**Total**		1499	100.0	2279	(100.0)	1184	(100.0)	

CRMHDP: Community based Registration and Management for Hypertension and type 2 Diabetes mellitus Project. ^1^: Proportion of more than 292 days covered in 365-days interval for two years. ^2^: Myocardial infarction, stroke, renal failure, glomerular disorder in diabetes, Arterial arteriolar and capillary disease, Diabetic retinopathy, Diabetic polyneuropathy, Diabetes with ulcer.

**Table 2 ijerph-18-03396-t002:** Difference of medication adherence rates between and within a paired group consisted of intervention group and control group over the period of one year before CRMHDP and two years after the project by Mixed model ANOVA (Analysis of Variance) adjusted for age, gender, household income status and comorbid complications of type2 diabetes mellitus.

	Source of Variation	Control A					Control B				
		Sum of Squares	DF	Mean Square	F	P	Sum of Squares	DF	Mean Square	F	P
CRMHDP-enrollees	Between Subject										
	Groups(G)	186.62	1	186.62	0.11	0.809	2519.56	1	2519.56	1.77	0.06
	Error	4,964,394	3711	1337.75			3,559,325	2671	1332.58		
	Within Subject										
	Time(T)	307,001.1	2	153,500.6	351.15	<0.001	236,386.6	2	118,193.3	298.73	<0.001 *
	Error	3,155,961	7422	425.22			2,124,914	5342	397.78		
	G * T	10,501.02	2	5250.51	13.31	<0.001 *	2077.32	2	1038.66	2.96	0.081

CRMHDP: Community based Registration and Management for Hypertension and type 2 Diabetes mellitus Project. *: It is statistically significant at the 95% confidence level.

**Table 3 ijerph-18-03396-t003:** Adjusted ratios of CRMHDP-enrollee group comparing to control groups in hospitalization due to type 2 diabetes-specific complications within five years after CRMHDP enrollment.

Adjusted Hazard Ratios * of CRHDP-Enrollee Group in Hospitalization Due to Diabetes Specific Complications (95% CI)	
Reference group: Control A area	0.76 (0.65–0.91)
Reference group: Control B area	0.52 (0.41–0.78)

* Adjusted variables: age, gender, economic status, type of health service security, and medication compliance during previous two years to participation. CRMHDP: Community based Registration and Management for Hypertension and type 2 Diabetes mellitus Project.

## Data Availability

This study was conducted using administrative data with the cooperation of the Korea National Health Insurance and the Korea Disease Control and Prevention Agency (NHIS-2020-1-323). This data is owned by the National Health Insurance and is accessible only to some researchers approved for personal information protection.

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
