# Peer review of "Effects of Community-Based Interventions on Medication Adherence and Hospitalization for Elderly Patients with Type 2 Diabetes at Primary Care Clinics in South Korea"

_ijerph, 2021, doi:10.3390/ijerph18073396_

Round 1
Reviewer 1 Report
The authors reported the medication adherence rates among those in CRMHDP more rapidly increased after the enrollment compared with the control groups and were maintained during the follow up period. The CRMHDP-enrollees were significantly less likely to be hospitalized by diabetes-specific complications than the control groups, after adjusting confounding factors.
However, some concerns have been raised.
- The study noted most of local physicians in the city participated. How many local physicians participated and how many local physicians did not participate in Gwangmyeong City and in area A and B? How many hospital numbers in the Gwangmyeong City and in area A and B? Did the doctors in the hospital participated in this program?
- How many people had DM in Gwangmyeong City and in area A and B? How many elder people had DM in Gwangmyeong City and in area A and B? How many elder people had DM in the hospital area in Gwangmyeong City and in area A and B?
- Is there any difference about medication adherence rates between CRMHDP and in hospital in Gwangmyeong City?
- Is there any sub-specifical doctor difference in this study in CRMHDP?
Author Response
Responses to the comments made by Reviewer 1
Many thanks for your comments. It was be very helpful to improve our manuscript.
Question 1) The study noted most of local physicians in the city participated. How many local physicians participated and how many local physicians did not participate in Gwangmyeong City and in area A and B? How many hospital numbers in the Gwangmyeong City and in area A and B? Did the doctors in the hospital participated in this program?
Response 1: This study included only local clinics but not hosptials in three areas. Since almost Korean local clinics were solo practices, we calculated the number of local clinics instead of physicians. We added the text regarding what you questioned in 148-171 lines, page 4.
Question 2) How many people had DM in Gwangmyeong City and in area A and B? How many elder people had DM in Gwangmyeong City and in area A and B? How many elder people had DM in the hospital area in Gwangmyeong City and in area A and B?
Response 2: We added the text regarding what you questioned in in 148-171 lines, page 4.
Question 3) Is there any difference about medication adherence rates between CRMHDP and in hospital in Gwangmyeong City?
Response 3: Since we excluded the patients treated at hospitals in the areas we could not compare medication adherence rates between patients at local clinics and hospitals. Given that the number of patients treated at hospitals in the local area were much fewer than one at local clinics, there might be not problem in excluding patients treated at hospitals in our study.
Question 4) Is there any sub-specifical doctor difference in this study in CRMHDP?
Response 4: We added the text regarding what you questioned in the lines 153-4 and 166-7, page 4.
Reviewer 2 Report
In this cohort study, a large sample size (intervention=1,132, control=1,512)) was a strength. The unique considerations of the Korean health system were well-articulated. Medication adherence was measured through medication prescription ratio (MPR), which appears to be measuring the dosage. It is not clear if this is a measurement of dosage prescribed or actual dosage used. Given the consideration of hospitalization due to complications, it appears to be measuring the actual dosage used.
Additional information that would be good to share include:
- Any health metrics used to confirm adherence
- Handling of confounding variables addressed, e.g., enhanced diabetes control due to weight loss
- Comparison of findings for the 20% who participated in the education sessions
Lines 33-34: Consider adding more current reference for the prevalence of DM
Figure 1: Recommend reworking lines and arrows to better indicate potential flow, some of the elements are not with any path
Figure 2: Recommend adding mean and/or range A1Cs for intervention and control groups?
Author Response
Responses to the comments made by Reviewer 2
Many thanks for your comments. It was be very helpful to improve our manuscript.
In this cohort study, a large sample size (intervention=1,132, control=1,512)) was a strength. The unique considerations of the Korean health system were well-articulated. Medication adherence was measured through medication prescription ratio (MPR), which appears to be measuring the dosage. It is not clear if this is a measurement of dosage prescribed or actual dosage used. Given the consideration of hospitalization due to complications, it appears to be measuring the actual dosage used.
Response : I agree on your opinion. As MPR was assessed in this study, we could not avoid to use dosage prescribed because we used Korean national health insurance claimed data. We added the text regarding your concern to limits of this study. Please see the lines 347-349, page 12.
Additional information that would be good to share include:
- Question 1) Any health metrics used to confirm adherence
Response 1: According to your suggestion, we made notion of MPR more clear. Plases see the lines 188-190, page 6.
- Question 2) Handling of confounding variables addressed, e.g., enhanced diabetes control due to weight loss
Response 2: Yes, we agree on your opinion. This study did not consider health behaviour like exercise and diet control as potential confouners. This words was added to limitations of this study. Please see the lines 338-340, page 12.
- Question 3) Comparison of findings for the 20% who participated in the education sessions
Response 3: Korean CRMHDP project focused on improving medication compliances and education program of the project need to be much more invested for effective program. We thought that details of participants in the education session did not need to be described in this manuscript
Question 4) Lines 33-34: Consider adding more current reference for the prevalence of DM
Response 4: According to your suggestion, it was updated to the current time. Please see the lines 1-3, page 1.
Question 5) Figure 1: Recommend reworking lines and arrows to better indicate potential flow, some of the elements are not with any path
Response 5) According to your suggestion, we fixed it. Please see Figure 1, page 4
Question 6) Figure 2: Recommend adding mean and/or range A1Cs for intervention and control groups?
Response 6) There was mismatch between figure 2 and your request. Please see that point.